# Quantitative Representation of Water Quality Biotoxicity by Algal Photosynthetic Inhibition

**DOI:** 10.3390/toxics11060493

**Published:** 2023-05-31

**Authors:** Li Hu, Tianhong Liang, Gaofang Yin, Nanjing Zhao

**Affiliations:** 1School of Physics and Material Engineering, Hefei Normal University, Hefei 230601, China; hulwuli@163.com; 2Key Laboratory of Environmental Optics and Technology, Anhui Institute of Optics and Fine Mechanics, Chinese Academy of Sciences, Hefei 230031, China; thliang@mail.ustc.edu.cn (T.L.); njzhao@aiofm.ac.cn (N.Z.)

**Keywords:** biotoxicity, algae, photosynthetic inhibition, multiple characterization parameters, data-driven method

## Abstract

The method based on the photosynthetic inhibition effect of algae offers the advantages of swift response and straightforward measurement. Nonetheless, this effect is influenced by both the environment and the state of the algae themselves. Additionally, a single parameter is vulnerable to uncertainties, rendering the measurement accuracy and stability inadequate. This paper employed currently utilized photosynthetic fluorescence parameters, including Fv/Fm(maximum photochemical quantum yield), Performance Indicator (PI_abs_), Comprehensive Parameter Index (CPI) and Performance Index of Comprehensive Toxicity Effect (PI_cte_), as quantitative toxicity characteristic parameters. The paper compared the univariate curve fitting results with the multivariate data-driven model results and investigated the effectiveness of Back Propagation(BP) Neural Network and Support Vector Machine for Regression (SVR) models to enhance the accuracy and stability of toxicity detection. Using *Dichlorophenyl Dimethylurea (DCMU)* samples as an example, the mean Relative Root Mean Square Error (RRMSE) corresponding to the optimal parameter PI_cte_ for the dose-effect curve fitting was 1.246 in the concentration range of 1.25–200 µg/L. On the other hand, the mean RRMSEs corresponding to the results of the BP neural network and SVR models were 0.506 and 0.474, respectively. Notably, BP neural network exhibited excellent prediction accuracy in the medium-high concentration range of 7.5–200 µg/L, with a mean RRSME of only 0.056. Regarding the stability of the results, the mean Relative Standard Deviation (RSD) of the univariate dose-effect curve results was 15.1% within the concentration range of 50–200 µg/L. In contrast, the mean RSDs for both BP neural network and SVR results were less than 5%. In the concentration range of 1.25–200 µg/L, the mean RSDs were 6.1% and 16.5%, with the BP neural network performing well. The experimental results of *Atrazine* were analyzed to further validate the effectiveness of the BP neural network in improving the accuracy and stability of results. These findings provided valuable insights for the development of biotoxicity detection by using the algae photosynthetic inhibition method.

## 1. Introduction

As unicellular organisms that play a significant role in ecosystems, the algae serve as excellent test organisms for accessing biological toxicity due to their sensitivity to toxins, short generation periods, ease of culture, and direct observation of toxicity symptoms at the cellular level [1]. The Organization for Economic Cooperation and Development (OECD) has recognized the algal growth inhibition test as a standard method for assessing the toxicity of chemicals [2]. Different from the growth inhibition method, which relies on cell density, cell yield, and growth rate as reaction endpoints, the algal photosynthetic inhibition method assesses the toxic effects of pollutants in water by measuring changes in photosynthetic fluorescence parameters based on rapid chlorophyll fluorescence induction kinetics. This method offers the advantages of rapid and convenient measurement and enables online detection in real time.

In recent years, experiments on the photosynthesis inhibition of algae by pollutant toxicity have been carried out extensively [3]. Strasser [4] derived a series of fluorescence parameters based on the theoretical models of biofilm energy flow. He used dose-effect (Log Logistic) curves to non-linear fit the inhibition rates of fluorescence parameters and pollutant concentrations and obtained dose-effect relationships between them. However, the quantitative analysis remained at the univariate curve fitting calibration stage. As research progressed, it was found that photosynthetic inhibition in algae was strongly influenced by environmental factors and the state of the algae themselves; relying on a single parameter resulted in uncertain measurements with lower accuracy and stability [5]. As a result, more parameters were needed to represent the photosynthesis inhibition effects, and quantitative analysis methods with multivariate inputs needed to be developed to reduce the interference of process factors by complementing information among variables. These improvements would lead to improved accuracy and stability of the algal photosynthesis inhibition method for biotoxicity detection.

## 2. Algorithm

Data-driven algorithms are well-suited for analyzing the complex relationship between dose and effect in algal photosynthesis inhibition detection. Such algorithms can effectively adjust parameters and extract the most relevant information about dependent variables, particularly when the relationship between multiple variables is unknown and difficult to discern. The commonly used data-driven models for this purpose are the Back Propagation(BP) neural network and Support Vector Regression(SVR).

### 2.1. BP Neural Network

The BP neural network is a multi-layer forward neural network based on the error backpropagation algorithm and has strong non-linear fitting and generalization capabilities. The network’s input forward propagation equation [6] is
(1)$$y_{h}=f_{1}\left(\underset{i}{\sum }w_{ih}I_{i}\right)$$
(2)$$C_{NNR}=f_{2}\left(\underset{h}{\sum }w_{hj}f_{1}\left(w_{ih}I_{i}\right)\right)$$

*C_NNR_* is the predicted output of the output layer, the pollutant concentration predicted by the neural network. $I_{i}$ represents the inhibition rate of photosynthetic fluorescence parameters after normalization, $y_{h}$ is the output of the hidden layer, $f_{1}$ is the S-shaped transfer function between the input layer and the hidden layer, $f_{2}$ is the linear transfer function between the hidden layer and the output layer, $W_{ih}$ is the connection weights between the input layer and the hidden layer, $W_{hj}$ is the connection weight between the hidden layer and the output layer. If the obtained output does not match the target output, the difference between the two is backward propagated, and the connection weights between layers are adjusted. The formulas for updating weights are
(3)$$w_{hj}\left(n+1\right)=w_{hj}\left(n\right)+\eta \delta_{j}y_{h}$$
(4)$$\delta_{j}=f_{2}^{\prime }\left(\underset{h}{\sum }(w_{hj}\left(n\right)y_{h}\right))\cdot \left(T_{j}-y_{j}\right)\cdot w_{ih}\left(n+1\right)$$
(5)$$w_{ih}\left(n+1\right)=w_{ih}\left(n\right)+\eta \delta_{h}y_{i}$$
(6)$$\delta_{h}=f_{1}^{\prime }\left(\underset{h}{\sum }(w_{ih}\left(n\right)I_{i}\right)\cdot \underset{j}{\sum }\left(w_{hj}\left(n\right)\delta_{j}\right))$$
$w_{hj}\left(n+1\right)$ and $w_{ih}\left(n+1\right)$ denote the ${\left(n+1\right)}_{th}$ weights, $w_{hj}\left(n\right)$ and $w_{ih}\left(n\right)$ denote the $n_{th}$ weights; $T_{j}$ is the $j_{th}$ node target value; η is the learning rate and takes the value in the range of 0 to 1. In this paper, we utilized Python 3.9 as the programming language and PyTorch 1.13.1 as the framework to train the BP neural network.

### 2.2. Support Vector Machine for Regression

SVR is an excellent algorithm for solving regression problems [7]. Based on statistical learning theory, this algorithm can find the best compromise between the complexity and the ability of the model on limited sample information; it can achieve the optimal learning effect while guaranteeing the generalization ability. The algorithm is especially suitable for handling small sample sizes and non-linear quantitative analysis [8] and can also handle highly correlated input variables. The formula of SVR is [9]
(7)$$C_{SVR}=\underset{i\in sv}{\sum }\alpha_{i}\cdot k_{rbf}\left(I_{i},I\right)+b^{*}$$

In the formula, $C_{SVR}$ is the pollutant concentration obtained from SVR, SV is the set of support vectors, $a_{i}$ and $l_{i}$ are the Lagrange Multiplier and support vectors, respectively. I is the input vector, b is a constant, $k_{rbf}$ is a radial basis function, the formulas are [9]:(8)$$k_{\mathrm{rbf}}(I_{i},I)=\exp(-\mathrm{gamma}{\left|I_{i}-I\right|}^{2})$$

The gamma is an important parameter in the kernel function. To improve the precision of predictions for the quantitative analysis of biotoxicity, the penalty factor C and gamma parameters need to be adjusted, and this paper employs the grid search method for parameter optimization. We used Python 3.9 as the programming language and SciKit-Learn 1.2.1 as the algorithm package in this paper, similar to the previous Neural Network section.

Multiple fluorescence parameters with high usage rates are extracted to compare the effectiveness of the input dose-effect curve fitting method with univariate input, the BP neural network, and SVR with multivariate input. This paper is going to investigate the effectiveness of the data-driven method with univariate input in improving the accuracy and stability of the algal photosynthetic inhibition method for biotoxicity detection.

## 3. Experiment Measurement and Parameters Selection

### 3.1. Experimental Measurement

The experiments in this paper used *Chlorella Pyrenoidosa*, a common algae species in freshwater, as the test organism and the pesticide *Dichlorophenyl Dimethylurea (DCMU)* and *Atrazine* as the stressors. The multi-phase chlorophyll fluorescence kinetic curves were measured by a variable light pulse-induced chlorophyll fluorescence analyzer (AGHJ-TPLIF-I, developed by Anhui Institute of Optics and Mechanics, Chinese Academy of Sciences) [10], as depicted in Figure 1. The selection and culturing methods were the same as in reference [11]. The initial chlorophyll concentration of the experimental algae was maintained at 100–200 µg/L, with 50 mL of algae liquid and 1 mL of stressor liquid (excluding the interference caused by *Dimethyl Sulfoxide* and dilution of the algae liquid), and the concentration and inhibition time of the stressors were set as indicated in Table 1. The measured fluorescence kinetic curves under *DCMU* and *Atrazine* stress were displayed in Figure 2, and the photosynthetic fluorescence parameters used in this paper were based on the inversion of these fluorescence kinetic curves.

### 3.2. Parameters Selection

The most used toxicity response parameters are Fv/Fm (maximum quantum yield of photosystem PSII) and the Performance Indicator for energy conservation from photons absorbed by PSII antenna to the reduction of QB (PI_abs_) [12,13,14,15]. Wang et al. [15] examined the stress response of *Microcystic aerugionosa* and *Pseudokirchneriella subcapitata* to *Tetracycline* and concluded that the maximum quantum yield of photosystem PSII (Fv/Fm) is a suitable indicator for *Tetracycline* toxicity detection. Li et al. [14] presented that Fv/Fm can be used to assess the toxicity of *Cu*^2+^. Strasser [4] constructed PI_abs_ based on a theoretical model of biofilm energy flow, considering the energy flow during photosynthesis. Sun et al. [16] demonstrated that PI_abs_ are the most convincing parameters among all fluorescence parameters by comparing the effects of *Atrazine* on different parameters of *Chlorella*. Based on the fact that the toxic stress caused the distortion of the algal fluorescence kinetic curve (OJIP), and the degree of change was proportional to the intensity of toxicity (as shown in Figure 2), some researchers extracted information on curve variability and obtained comprehensive fluorescence parameters to characterize the biological toxicity of water bodies, such as Comprehensive Parameter Index (CPI) and Performance Index of Comprehensive Toxicity Effect (PI_cte_). Moreover, they proved that the comprehensive parameters were superior in terms of toxicity response, minimum detection limit, maximum response concentration, stability, and reproducibility under stress with *DCMU*, *Dibromothymoquinone* (*DBMIB), Methyl Viologen (MV), Malathion* and *Carbofuran* [17,18].

In this paper, based on the above research about toxicity parameters, the photosynthetic fluorescence parameters Fv/Fm and performance indicators PI_abs_, comprehensive parameters CPI and PI_cte_, would be extracted and used as input parameters for univariate dose–effect curve fitting and multivariate BP neural network and SVR.

## 4. Method and Results

As explained in Section 3.2, we conducted dose-effect curve, BP neural network, and SVR analyses using normalized Fv/Fm, PI_abs_, CPI and PI_cte_ as independent variables, while the sample concentration served as the dependent variable. We established n experimental concentrations and used n-1 of them as the training set and 1 as the test set to obtain predicted results for all samples. We evaluated the accuracy of predictions using the Relative Root Mean Square Error (RRMSE) and the stability using the Relative Standard Deviation (RSD) [19].
(9)$$RRMSE=\frac{1}{x_{t}}\sqrt{\frac{\sum_{i=1}^{n}(x_{i}-x_{t}{)}^{2}}{n}}$$

In the formula, $x_{i}$ is the prediction corresponding to a sample, $x_{t}$ is the true concentration, *n* is the times of repeat experiments.
(10)$$RSD=\left(\frac{1}{\bar{x}}\sqrt{\frac{\sum_{i=1}^{n}x_{i}-\bar{x}}{n-1}}\right)\times 100\%$$

In the formula, $x_{i}$ is the prediction corresponding to a sample, $\bar{x}$ is the mean concentration, *n* is the times of repeat experiments.

### 4.1. Analysis of Single Parameter for Dose–Effect Curve Fitting

Initially, the inhibition rates of Fv/Fm, PI_abs_, CPI and PI_cte_ variables were set in a dose–effect curve fitting independently to predict concentration. The results presented that RRMSEs of samples in concentrations of 0.625 µg/L and 0.75 µg/L were greater than 10 and thus not significantly reliable. Only results in the effective concentrations range of 1.25–200 µg/L were included below. Comparing the accuracy of the prediction results for different parameters, RRMSEs corresponding to Fv/Fm, PI_abs_, CPI and PI_cte_ were 2.25, 1.31, 1.54 and 1.25, respectively. The three comprehensive parameters predicted higher accuracy than Fv/Fm alone, with PI_cte_ having the lowest RRMSE and a 44.9% decrease compared to Fv/Fm. This suggested that the comprehensive parameters reflect the effects of multiple nodes, rather than one node of the photosynthesis process, and provide more valid information and greater accuracy in predicting samples of unknown concentrations. Further analysis of the trend of the RRMSE for each parameter with increasing concentration showed that the RRMSE of the three comprehensive parameters rapidly decreased with increasing concentration and stabilized in the middle and high range. However, the accuracy and the stability of predictions were not high enough, as shown in Figure 3a.

Regarding stability, multiple parallel samples in the low to medium concertation region (0.625–32 µg/L) had a consistent RSD of zero, and this was not in line with reality. The sensitivity to changes in independent variables decreased sharply for the larger values of the power *p* of the Logistic function ($y=A_{2}+\left(A_{1}-A_{2}\right)/(1+{\left(x/x_{0}\right)}^{p}$) corresponding to the dose-effect curve, which seriously affected the sensitivity of the detection in the low-concentration region. Therefore, only results in the concentration range of 50–200 µg/L were considered. The RSDs corresponding to Fv/Fm, PI_abs_, CPI and PI_cte_ were 35.4%, 19.9%, 17.2% and 15.1%, respectively, while the PI_cte_ parameter had the smallest RSD, which was reduced by 57.3% relative to Fv/Fm, as shown in Figure 3b.

### 4.2. Analysis of Multi-Parameters for Data-Driven Model

The normalized inhibition rates of Fv/Fm, PI_abs_, CPI and PI_cte_ were used as the independent variables for regression analysis in the data-driven model, and the concentration of samples was used as the dependent variable. Univariate dose–effect curves for PI_cte_ were compared to multivariate input predictions using BP neural network and SVR, as indicated in Table 2.

The mean RRMSE values for the effective concentration range (1.25–200 µg/L) were 1.246, 0.506, and 0.474 for the dose–effect curve, BP neural network, and SVR, respectively. The BP neural network and SVR models clearly outperformed the conventional dose–effect curve, with a reduction in RRMSE values of 59.3% and 61.9%, respectively. An analysis of the RRMSE trends with concentration indicated that the data-driven model predictions decreased rapidly with increasing concentration and were stable in medium- to high-concentration regions, as depicted in Figure 4a. The mean RRMSE of the BP neural network in the medium to high concentration range (7.5–200 µg/L) was only 0.056, demonstrating high accuracy, attributed to its strong non-linear fitting capability in approximating any non-linear continuous function with arbitrary precision.

Comparative analysis of the stability of the prediction results revealed that the mean RSD of the univariate dose–effect curve fitting for PI_cte_ was 15.1% in the effective concentration interval (50–200 µg/L), while both data-driven models showed mean RSDs of less than 5% in this concentration region. For the effective concentration interval (1.25–200 µg/L), the RSDs of the multivariate BP neural network and SVR predictions decreased with increasing concentration, with mean values of 6.1% and 16.5%, respectively, and the BP neural network predictions were more stable.

In summary, for *DCMU* samples, BP neural network with multivariate inputs exhibited superior predictive accuracy and stability relative to dose–effect curve fitting and SVR, owing to their strong non-linear fitting capability.

### 4.3. Experimental Verification of the Other Substance

The methodology was applied to Atrazine samples to assess the efficacy of multivariate BP neural network in improving prediction accuracy. Initially, dose-effect curve fitting was conducted with Fv/Fm, PI_abs_, CPI and PI_cte_ inhibition rate used as independent variables, and with the concentration of the samples used as the dependent variable. The mean RRMSEs in the effective concentration range (1.25–200 µg/L) were obtained for each parameter and were 2.20, 1.59, 1.40 and 1.40, respectively. The predictions of the three comprehensive parameters were found to be more accurate than the common parameter Fv/Fm, and there were the most accurate results for CPI and PI_cte_. The mean RSDs corresponding to the parameters Fv/Fm, PI_abs_, CPI and PI_cte_ in the effective concentration interval (50–200 µg/L) were 16.4%, 16.7%, 19.7% and 17.0%, respectively. There was no significant difference in the RSDs of the predicted results for the four parameters using a one-ANOVA analysis of variance.

Figure 5 displayed a further comparison of the multivariate BP neural network and SVR predictions. There was a higher accuracy for BP neural network and SVR predictions, with mean RRMSE values of 0.532 and 1.379, respectively. The accuracy of BP neural network prediction was the highest, and the RRMSEs were decreased by 62.0% relative to the results of the univariate PI_cte_ for dose-effect fitting. In the whole concentration region, the RRMSEs of the BP neural network predictions remained basically unchanged with concentration, demonstrating good accuracy. Additionally, in the effective concentration region (50–200 µg/L), the mean RSD of the BP neural network predictions was 9.8%, lower than the dose-effect curve fitting results. These results were consistent with those obtained from *DCMU* samples in Section 4.2, which further illustrated the ability of BP neural networks to significantly improve the accuracy and stability of toxicity detection of photosynthetic inhibition effects in algae.

## 5. Conclusions

In this paper, we examined the effectiveness of a multivariate BP neural network in improving the accuracy of predictions using four commonly used photosynthetic fluorescence parameters, namely Fv/Fm, Pl_abs_, CPI and PI_cte_. We compared and analyzed the accuracy and stability of the inversion results of the univariate dose–effect curve and multivariate BP neural network and SVR. Firstly, we analyzed the results of the four parameters entered individually into the dose–effect curve for *DCMU* samples. The mean RRMSE and RSD values corresponding to the comprehensive parameter PI_cte_ were 1.26 (1.25–200 µg/L) and 15.1% (50–200 µg/L), respectively, which were better than those of the other parameters but not sufficiently high. We then compared the results of the BP neural network and SVR with multivariate input. The corresponding mean RRMSEs were 0.506 and 0.474, respectively, with a mean RRMSE of only 0.056 in the medium to high concentration range (7.5–200 µg/L) for the BP neural network. The BP neural network also exhibited good stability, with an average RSD of only 6.1% (1.25–200 µg/L). Finally, the experimental finding of *Atrazine* samples further verified that the BP neural network could effectively improve the accuracy and stability of algal photosynthesis inhibition methods. In the next step, the BP neural network quantitative analysis will be optimized for the practical situation of the algal photosynthetic inhibition method on biotoxicity detection.

## Figures and Tables

**Figure 1 toxics-11-00493-f001:**
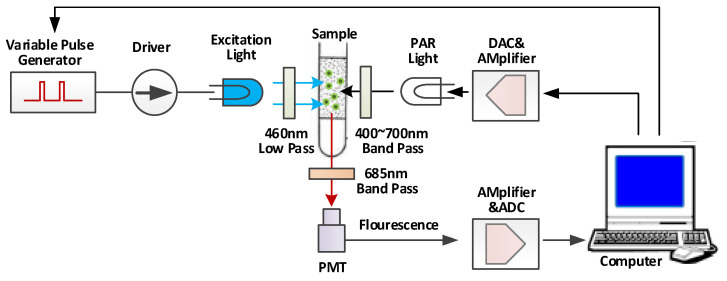
Schematic diagram of chlorophyll fluorescence analyzer induced by variable light pulse.

**Figure 2 toxics-11-00493-f002:**
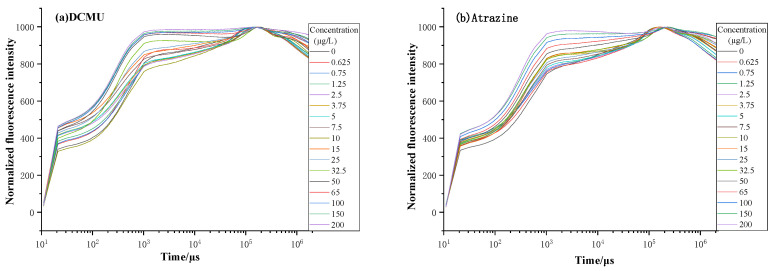
Fluorescence kinetic curve under stress of (**a**) *DCMU*; (**b**) *Atrazine*.

**Figure 3 toxics-11-00493-f003:**
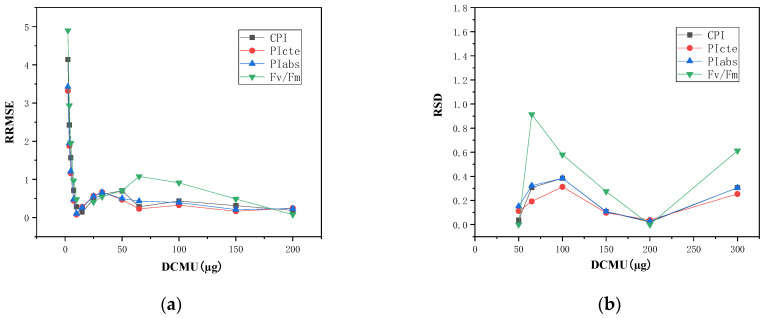
Predictions of *DCMU* by dose–effect curve-fitting (**a**) RRMSEs trend with concentration (**b**) RSDs trend with concentration.

**Figure 4 toxics-11-00493-f004:**
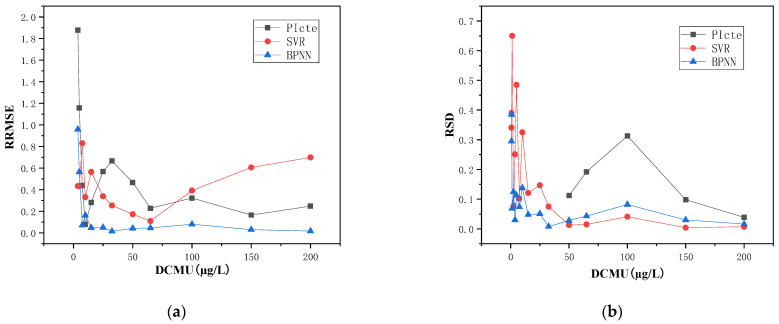
Predictions of DCMU by data-driven models (**a**) RRMSEs trend of concentration (**b**) RSDs trend of concentration. (Note: BPNN stands for BP neural network).

**Figure 5 toxics-11-00493-f005:**
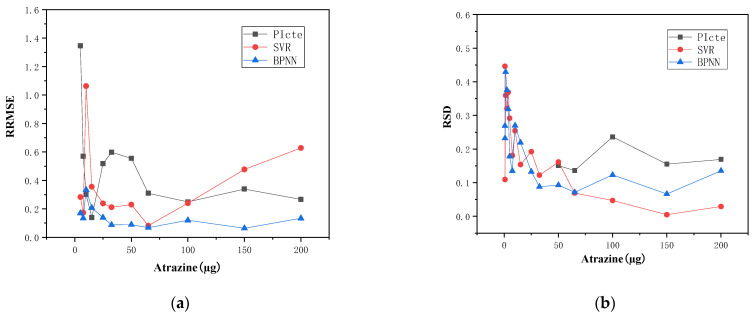
Predictions of *Atrazine* by data-driven models (**a**) RRMSEs trend of concentration (**b**) RSDs trend of concentration.

**Table 1 toxics-11-00493-t001:** Concentration and stress time of two toxic substances to be measured in algal fluid.

Toxin	Concentrations Tested	Stress Time
** *DCMU* **	0.625, 0.75, 1.25, 1.5, 2.5, 3.75, 5, 7.5, 10, 15, 25, 32.5, 50, 75, 100, 150, 200 μg /L	15 min
** *Atrazine* **	0.625, 0.75, 1.25, 1.5, 2.5, 3.75, 5, 7.5, 10, 15, 25, 32.5, 50, 75, 100, 150, 200 μg/L	15 min

**Table 2 toxics-11-00493-t002:** Predictions of *DCMU* by Data-driven Models.

Sample Groups	Concentration µg/L	RRMSE			RSD		
		Dose–Effect Curve	SVR	BP Neural Network	Dose–Effect Curve	SVR	BP Neural Network
1	0.625	16.266	9.923	8.867	--	0.341	0.295
2	0.75	13.389	4.913	5.318	--	0.390	0.385
3	1.25	7.633	0.875	2.518	--	0.650	0.069
4	2.5	3.317	0.596	2.483	--	0.078	0.125
5	3.75	1.878	0.435	0.959	--	0.251	0.030
6	5	1.158	0.432	0.564	--	0.485	0.113
7	7.5	0.439	0.831	0.072	-	0.102	0.074
8	10	0.079	0.331	0.162	--	0.325	0.138
9	15	0.281	0.565	0.048	--	0.121	0.048
10	25	0.568	0.339	0.050	--	0.147	0.051
11	32.5	0.667	0.254	0.015	0.002	0.075	0.008
12	50	0.467	0.174	0.042	0.112	0.013	0.028
13	65	0.228	0.110	0.046	0.192	0.015	0.043
14	100	0.323	0.392	0.080	0.313	0.041	0.082
15	150	0.166	0.606	0.030	0.098	0.004	0.030
16	200	0.248	0.699	0.017	0.039	0.007	0.016
Average (µg/L)		1.246	0.474	0.506	0.151	0.165	0.061

## Data Availability

For personal reasons, the data will not be disclosed for the time being.

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
