# Peer review of "Quantitative Representation of Water Quality Biotoxicity by Algal Photosynthetic Inhibition"

_toxics, 2023, doi:10.3390/toxics11060493_

Round 1

Reviewer 1 Report

The article employed utilized photosynthetic fluorescence parameters as quantitative toxicity characteristic parameters via compared the univariate curve fitting results with the multivariate data-driven model results and investigated the effectiveness of back-propagation neural network and Support Vector Machine for Regression models. The chosen topic is very important in quantitative representation of water quality biotoxicity. However, my attention has been addressed to minor aspects. In addition to the writing's politeness, I kindly recommend that the following be corrected:

-     In the Abstract: Please write the full name before the abbreviation such as PIabs, CPI, PIcte, DCMU and SVR.

-     In the manuscript: Please change ug/L to mg/L (lines 17, 20, 22, 23, 111, 119 Table 1, 160, ---etc.)

-     Line 44: Please follow the author (Strasser) with the reference number.

-     After Line 79: Please provide the software name and version of the BP Neural Network program that was used. Please include any relevant program parameters such as normalization equation of the inputs, number of hidden layers, activation function and de-normalization equation of the output. In results section, I recommend to add Table for weighs and biases values between input and hidden layers.

-     After Line 101: Please provide the software name and version of the SVR program that was used.

-     In Figure 2: Please check Time/us. Do you mean ms?

-     Line 124 and 130: Please subscript abs in PIabs.

-     Line 124: Please change Wang [15] to Wang et al. [15].

-     Line 127: Do you mean Yang et al. [16] or Li et al. [14]?

-     Line 127: Please superscript 2+ in Cu2+.

-     Line 129: Please change Sun [17] to Sun et al. [17].

-     Line 136: Please subscript cte in PIcte.

-     Line 268: Please include the title: "Water quality-freshwater algal growth inhibition test with unicellular green algae" in the reference.

-     Line 272: Please correct the reference to be: Strasser, R.J., Tsimilli-Michael, M., Srivastava, A. (2004). Analysis of the Chlorophyll a Fluorescence Transient. In: Papageorgiou, G.C., Govindjee (eds) Chlorophyll a Fluorescence. Advances in Photosynthesis and Respiration, vol 19. Springer, Dordrecht. https://doi.org/10.1007/978-1-4020-3218-9_12

Minor editing of English language required.

Author Response

Thank you very much for your review and professional opinions, which have been revised one by one according to the revised opinions. Please also check the attachment, which contains the expert opinion reply.

Best wishes and looking forward to your reply.

Li Hu

Hefei Normal University

Reviewer 2 Report

Title: Quantitative Representation of Water Quality Biotoxicity by Algal Photosynthetic Inhibition

Authors: Li Hu, Tianhong Liang, Gaofang Yin, and Nanjing Zhao
Manuscript ID: toxics-2389628

General Comments:

This manuscript includes an interesting work to improve a biotoxicity detection by employing BP neural network and SVR models using multivariate photosynthetic fluorescence parameters of algae. However, authors need to improve the manuscript before the publication. First of all, authors need to clarify what is their goal and what they achieved through their work in the conclusions section considering the goal.

Specific Comments:

Abstract

 Abbreviation of terms should be explained when the terms are mentioned first unless they are commonly used.

Line 17, 22, 23: u? micro? Please check the unit.

1. Introduction

It is not clear what is the goal of this study. Authors need to clarify their study goal.

 2. Algorithm

Line 61-62: It is time to insert the abbreviation BP and SVR here.

Line 68-76, 89-92: It is hard to follow the description since the explanation for symbols was not in order.

Line 93: What is RBF?

 3. Experiment Measurement & Parameters Selection

Line 102: The species name should be italic and this applies for other species names throughout the text.

Line 111: ug? Please check the unit throughout the text including tables (e.g. Table 1) and correct them.

Line 123, 126: Explanation for Fv/Fm should be in line 123.

Line 124-125: Please check the species name in italic.

Line 127: Cu2+? Please check the typo.

4. Results and Discussion

Line 146-156: The paragraphs include methods not results or discussion.

Line 162: blow?

Table 2: What is the line below 1 in the column of sample groups?

Line 251: cure?

Author Response

(The authors gave the same response as above.)

Round 2

Reviewer 1 Report

Thank you for the authors' responses. All of my comments were addressed by the authors, who also changed most of them in the final version of the manuscript. Please include the response to the comment regarding the name and version of the BP Neural Network and SVR programs used in the manuscript. Please begin "2.2. Support Vector Machine for Regression" in a new line at line 82 of the revised manuscript.  

Author Response

(The authors gave the same response as above.)
